

# Lysyl oxidase-like 1 predicts the prognosis of patients with primary glioblastoma and promotes tumor invasion *via* EMT pathway

Gui-Qiang Yuan[1,2], Guoguo Zhang[2], Qianqian Nie[2], Zhong Wang[2], Hong-Zhi Gao[3], Gui-Shan Jin[1] and Zong-Qing Zheng[3,4]

[1] Beijing Neurosurgical Institute, Department of Neurosurgery, Beijing Tiantan Hospital Affiliated to Capital Medical University, Capital Medical University, Beijing, China
[2] Department of Neurosurgery & Brain and Nerve Research Laboratory, The First Affiliated Hospital of Soochow University, Suzhou, Jiangsu, China
[3] Department of Neurosurgery, The Second Affiliated Hospital of Fujian Medical University, Quanzhou, Fujian, China
[4] Department of Neurosurgery, Neurosurgery Research Institute & Binhai Branch of National Regional Medical Center, The First Affiliated Hospital, Fujian Medical University, Fujian, Fuzhou, China

Corresponding authors
Gui-Shan Jin, guishanjin7@163.com
Zong-Qing Zheng,
zongqing2007@126.com

## ABSTRACT

**Background:** Lysyl oxidase enzymes (LOXs), as extracellular matrix (ECM) protein regulators, play vital roles in tumor progression by remodeling the tumor microenvironment. However, their roles in glioblastoma (GBM) have not been fully elucidated.

**Methods:** The genetic alterations and prognostic value of LOXs were investigated *via* cBioPortal. The correlations between LOXs and biological functions/molecular tumor subtypes were explored in The Cancer Genome Atlas (TCGA) and the Chinese Glioma Genome Atlas (CGGA). After Kaplan–Meier and Cox survival analyses, a Loxl1-based nomogram and prognostic risk score model (PRSM) were constructed and evaluated by time-dependent receiver operating characteristic curves, calibration curves, and decision curve analyses. Tumor enrichment pathways and immune infiltrates were explored by single-cell RNA sequencing and TIMER. Loxl1-related changes in tumor viability/proliferation and invasion were further validated by CCK-8, western blot, wound healing, and Transwell invasion assays.

**Results:** GBM patients with altered LOXs had poor survival. Upregulated LOXs were found in IDH1-wildtype and mesenchymal (not Loxl1) GBM subtypes, promoting ECM receptor interactions in GBM. The Loxl1-based nomogram and the PRSM showed high accuracy, reliability, and net clinical benefits. Loxl1 expression was related to tumor invasion and immune infiltration (B cells, neutrophils, and dendritic cells). Loxl1 knockdown suppressed GBM cell proliferation and invasion by inhibiting the EMT pathway (through the downregulation of N-cadherin/Vimentin/Snai1 and the upregulation of E-cadherin).

**Conclusion:** The Loxl1-based nomogram and PRSM were stable and individualized for assessing GBM patient prognosis, and the invasive role of Loxl1 could provide a promising therapeutic strategy.

# INTRODUCTION

Glioblastoma (GBM) is a formidable lethal malignancy that affects 21% of the glioma patient population and contributes to 7% of all brain tumors. This pathology is notably typified by a heightened propensity for both recurrent manifestations and a high incidence of functional impairment (*Ostrom et al., 2017*). Although sufficient surgical tumor resection, radiotherapy, and chemotherapy are used (*Stupp et al., 2005*), the prognosis of GBM patients remains dismal, with a survival rate of only 4–7% (*Ostrom et al., 2022*). Currently, numerous novel biomarkers have been identified that exhibit potential prognostic value. Nonetheless, these indicators fail to fully demonstrate the individual prognosis due to GBM heterogeneity and asymmetrical clinical features. Thus, developing an effective and individualized tool for assessing GBM patient prognosis is critical and may contribute to clinical practice.

Lysyl oxidase enzymes (LOXs) include Lox and lysyl oxidase-like proteins 1–4 (Loxl1-4), which are encoded by chromosomes 5q23.1, 15q24.1, 8p21.3, 2p13.1 and 10q24.2 (*Asuncion et al., 2001*). These five members function as secretory copper-dependent amine oxidases due to their highly conserved C-terminal domains, including copper binding motifs and lysyl-tyrosyl-quinone (LTQ) cofactors (*Wang, Hsia & Shieh, 2016*). In addition, their diverse N-terminal domains further divide LOXs into two subfamilies: Lox, Loxl1 (immature pro-domians) and Loxl2-4 (scavenger-receptor cysteine-rich (SRCR) domains) (*Xiao & Ge, 2012*). The former needs bone morphogenic protein-1 (BMP-1) protease to transform pro-Lox and pro-Loxl-1 into maturation, which Loxl2-4 need not (*Borel et al., 2001*). LOXs mainly promote covalent cross-linking between collagens and elastin in the extracellular matrix (ECM) and strengthen the structural stress and integrity of tissues (*Wullkopf et al., 2018*). The ECM regulates cellular transformation and metastasis to promote tumor progression (*Seewaldt, 2014*), which is also a major component of the tumor microenvironment (TME). LOXs secreted from cells participate in the regulation of the TME and can be absorbed or returned to the cytoplasm to influence cell phenotypes (*Barker, Cox & Erler, 2012*). Mutations and upregulation of LOXs have been found in many cancers. In glioma research, LOXs expression can aggravate the malignant phenotypes of tumor cells (*Laurentino et al., 2022*). For example, Lox is upregulated in glioma (*da Silva et al., 2015*), and knockdown or inhibition of Lox can compromise tumor migration, invasion, and angiogenesis (*Kim et al., 2014*; *Kore et al., 2018*; *Mammoto et al., 2013*). Lox-related pathways were found in hypoxia-regulated GBM exosomes (*Kore et al., 2018*; *Kucharzewska et al., 2013*). Lox also participates in the progression of macrophage infiltration (*Chen et al., 2019*; *Zhao et al., 2021a*). Loxl2 can activate glioma cell autophagy to induce the epithelial-to-mesenchymal transition (EMT) process and reduce chemotherapy sensitivity (*Zhang et al., 2020*). Low Loxl3 decreased glioma invasion and restrained tumor progression (*Laurentino et al., 2021*). High Loxl1 is related to glioma cell apoptosis and migration *via* Wnt/beta-catenin signaling or stabilizing BAG2 (*Li et al., 2018*; *Yu et al., 2020*). However, the role of Loxl1 is still unclear in glioma cell invasion.

Through single-cell sequencing analysis, the roles of Loxl1 in the TME of glioma could be further explored. In addition, the prognostic role of LOXs family in primary GBM patients has not been fully investigated, and the clinical application of LOXs-based prediction models could also be further explored.

Our present study comprehensively investigated the prognostic value of LOXs at the genomic and transcriptional levels in GBM. The biological functions of LOXs in GBM were determined by GO and KEGG analyses. The categorization of GBM subtypes in diverse patient cohorts based on distinct levels of LOXs was examined utilizing data from The Cancer Genome Atlas (TCGA), with subsequent validation conducted within the Chinese Glioma Genome Atlas (CGGA) repository. Survival analyses were performed to screen for independent clinicopathological factors associated with GBM prognosis. Subsequently, the validation cohort included diverse GBM cohorts, culminating in the formulation of the nomogram and a prognostic risk score model (PRSM) centered on Loxl1. Furthermore, an in-depth exploration of the expression patterns of Loxl1 within GBM was undertaken through single-cell RNA sequencing (scRNA-seq) analyses. Loxl1-related pathways and immune infiltration in GBM were also explored by gene set enrichment analysis (GSEA), gene set variation analysis (GSVA), and TIMER. To elucidate the contribution of Loxl1 to the progression of GBM, an array of investigative methodologies, including the CCK-8 assay, western blot analysis, wound healing assays, and transwell invasion assays, were used. Comprehensive analyses of the LOXs family, especially Loxl1, could provide prognostic and therapeutic targets for GBM patients.

# MATERIALS AND METHODS

## Prognostic variants identified *via* the genomic sequencing of LOXs in GBM

cBioPortal integrates gene expression, alteration information, and clinical data from multiple databases (*Gao et al., 2013*). We integrated 429 GBM samples with complete mutation and copy number alteration (CNA) data (Table S1: Brain Tumor PDXs, Mayo Clinic (*Vaubel et al., 2020*); Glioblastoma, Columbia (*Zhao et al., 2019*); GBM, TCGA, PanCancer Atlas) to analyze the ratio and sites of each LOXs genetic alteration. The potential prognostic value of LOXs genetic alterations was also investigated with Kaplan–Meier survival analysis.

## TCGA and CGGA databases

The GBM cohorts included both isocitrate dehydrogenase (IDH)1-wild-type (WT) and IDH1-mutant patients from GlioVis (TCGA) (*Bowman et al., 2017*) and the CGGA website (*Zhao et al., 2021b*). We explored the potential diagnostic and prognostic value of LOXs *via* differential expression and survival analyses. After constructing the nomogram and PRSM, we further included another TCGA GBM cohort to test the PRSM.

## Biological function analysis of LOXs

A total of 250 genes related to LOXs (50 of each member) in GBM based on the Pearson correlation coefficient were obtained from GEPIA (*Tang et al., 2019*). Based on the

integrated groups of LOXs-related genes, a protein–protein interaction network was constructed (Table S2) *via* STRING (*von Mering et al., 2005*). Further GO and KEGG biological function analyses were subsequently conducted, and the results were visualized with Cytoscape (version 3.8.2).

## Survival analysis, model construction, and evaluation

The prognostic role of each LOXs family member in GBM was analyzed by Kaplan–Meier analysis in GBM cohorts (TCGA training cohort, $n = 330$; Table S3; CGGA validation cohort, $n = 193$; Table S4; and external TCGA validation cohort, $n = 266$; Table S5). Subsequently, the prognostic value of LOXs was explored through Cox survival analyses, which revealed clinicopathological variables such as age, sex, tumor classifications (including IDH mutation status and GBM subtypes), and treatment modalities (including chemotherapy and radiotherapy). Based on these discerned prognostic factors, a nomogram and the PRSM were systematically constructed, providing a robust framework for appraising the 1- and 2-year prognostic outcomes for GBM patients. The predictive risk score formula was as follows: Y = 0.115*Loxl1 + 0.362*Age-0.739*IDH1-1.010*Chemo-1.824*Radio. After excluding GBM patients with IDH1 mutations according to the 2021 WHO CNS tumor classification (*Louis et al., 2021*), the training TCGA (IDH1-WT, $n = 309$), CGGA (IDH1-WT, $n = 161$), and external TCGA (IDH1-WT, $n = 246$) GBM datasets were used to further assess the model. Time-dependent receiver operating characteristic (t-ROC) curves were used to analyze the model prediction accuracy. The net clinical benefits were judged by calibration and decision curve analyses. The entire spectrum of the data was subjected to rigorous analysis and subsequent visualization, a process meticulously executed through the use of R (version 4.0.3).

## Single-cell RNA sequencing data processing

The GSE182109 datasheet (*Abdelfattah et al., 2022*) from the GEO database contained low-grade glioma (LGG) and newly diagnosed GBM (ndGBM, primary GBM) groups. The quality control metrics are listed below, and the following exclusion criteria were as follows: genes detected in <3 cells, <300 genes detected in cells, a proportion of mitochondria >10%, and a percentage of hemoglobin genes >3%. A total of 3,000 highly variable feature genes were marked by the function "FindVariableFeatures" from the R package "Seurat". The quality-controlled data were scaled by the function "ScaleData", and the data dimensionality was reduced by the function "RunPCA", according to highly variable genes. Finally, the cells were clustered by the functions "FindNeighbors" and "FindClusters" (Dim = 20, Solution = 0.4). T-distributed stochastic neighbor embedding (t-SNE) was used to further reduce the data dimensions and visualize the distribution on the two-dimensional plot. As described in a previous study of GSE182109, marker genes for cell type annotation were obtained and are listed in Table S6 (*Abdelfattah et al., 2022*). We visualized the expression pattern of each marker gene by the function "DotPlot" and annotated the cell types.

## Functional gene enrichment analyses based on scRNA-seq

To identify the enriched functions between the LGG and GBM subgroups, gene set enrichment analysis (GSEA) was conducted with the Molecular Signatures Database (MSigDB) (c2.cp.kegg.v2023.1.Hs.symbols.gmt) *via* the R package (*Liberzon et al., 2015*). Gene set variation analysis (GSVA) was also performed to calculate the pathway scores of each cell using the R package "GSVA". The correlation between Loxl1 and enriched biological processes in GBM could be further explored.

## Analysis of tumor-infiltrating immune cells

The correlations between Loxl1 and tumor-infiltrating immune cells (B cells, CD8+ T cells, CD4+ T cells, macrophages, neutrophils, and dendritic cells) in GBM were explored and visualized with TIMER (https://cistrome.shinyapps.io/timer/) (*Li et al., 2017*).

## Cell culture and construction of the Loxl1 knockdown cell line

The U87 MG human glioblastoma cell line was acquired from the American Type Culture Collection (Virginia, USA) and cultured in Dulbecco's modified Eagle's medium (DMEM) (Gibco, Billings, MT, USA) supplemented with 10% fetal bovine serum (FBS, F8318-500ML) in a 37 °C and 5% $CO_2$ humidified incubator.

Loxl1 knockdown lentiviral vectors containing Loxl1-specific small hairpin RNA (shLoxl1-1: CCTGGGAACTACATCCTCA, shLoxl1-2: GCATTAAAGCAGCGTATC) and nontargeting shRNA (shNT: CTCGCTTGGGCGAGAGTAA) were purchased from GeneChem (Shanghai, China). In accordance with the manufacturer's instructions, U87 MG cells were cultured at a density of $10 \times 10^4$ cells/well in a six-well plate before transfection. The medium was changed to serum-free DMEM. Then, shNT or shLoxl1-1&2 lentivirus was separately added to the wells at an MOI of 1.5. After 6 h, the lentivirus medium was discarded, and the medium was supplemented with puromycin (1.5 μg/ml). The drug selection medium was replaced with complete medium 24 h later, and Loxl1-knockdown cell lines were constructed for further experiments.

## Cell viability and proliferation assay

A cell counting kit-8 (CCK-8) assay was used to detect the effect of Loxl1 knockdown on the viability and proliferation of U87 MG cells. A total of 2,500 stable knockdown cells per well were seeded in a 96-well plate. We measured the absorbance of each well every 24 h for 96 h. Ten microliters of CCK-8 reagent (521942; Biosharp, Heifei, China) was added to each well, and the plate was placed in an incubator at 37 °C for 1 h. The absorbance of the different groups at 450 and 670 nm was read with a microplate reader. Cell viability and proliferation curves were generated and analyzed with GraphPad Prism and SPSS. Multiple comparisons were conducted between the shNT group and the sh-Loxl1-1 & 2 group at each time point.

## Western blot analysis

Transfected U87 MG cells were cultured in six-well plates and harvested when they reached >80% confluence. Each well was washed with PBS twice and lysed with RIPA

buffer (P0013B; Beyotime, Shanghai, China) supplemented with PMSF for 15 min on ice. Then, we scratched off the cells with scrapers and centrifuged them for 20 min at 4 °C. The supernatant was carefully collected, quantified with a bicinchoninic acid (BCA) kit (P0011; Beyotime, Shanghai, China) and diluted to 1 mg/ml with RIPA buffer. Then, the loading buffer was mixed with the supernatant, and the mixture was denatured at 100 °C for 10 min. The denatured samples were separated by sodium dodecyl sulfate–polyacrylamide gel electrophoresis (120 V, 60 min) and transferred to a nitrocellulose filter membrane (Millipore, Burlington, MA, USA), whose weight was based on the protein molecular weight. After blocking with 5% BSA at room temperature for 1 h, the membrane was incubated with primary antibodies (anti-Loxl1 (sc-166632, Santa Cruz, CA, USA), anti-N-cadherin (#13116; Cell Signaling Technology, Danvers, MA, USA), anti-E-cadherin (#3195; Cell Signaling Technology, Danvers, MA, USA), anti-Vimentin (#5741; Cell Signaling Technology, Danvers, MA, USA), anti-Snai1 (13099-1-AP; Proteintech, Rosemont, IL, USA), anti-GAPDH (60004-1-Ig; Proteintech, Rosemont, IL, USA), and anti-β-actin (20536-1-AP; Proteintech, Rosemont, IL, USA)) on a shaker at 4 °C overnight, followed by incubation with secondary antibodies (anti-mouse, SA00001-1 & anti-rabbit, SA00001-2; Proteintech, Rosemont, IL, USA) for 1 h at room temperature. Enhanced chemiluminescence reagent (P0018S; Beyotime, Shanghai, China) was added to visualize the bands with a ChemiDoc MP (Bio-Rad, Hercules, CA, USA). The gel bands were quantified by ImageJ (NIH, Bethesda, MD, USA), and the GAPDH/β-actin ratio was used as an internal control. The target protein/internal control in the shNT group was compared with the ratios in the Loxl1 knockdown groups.

## Wound healing assay

U87 MG cells were subjected to different groups in six-well plates and reached >90% confluence. Then, a 10-μl pipette tip was used to scratch the bottom, and the medium was changed to serum-free DMEM. The wound gaps were photographed at 0 and 24 h at the same location. The wound gaps were measured by ImageJ software, and the ratio of the closure gap was analyzed between the shNT group and the sh-Loxl1-1&2 groups to detect differences in invasion ability.

## Transwell invasion assay

U87 MG cells from different groups were counted and separately cultured in Transwell chambers (8.0 μm pore) at $1.5 \times 10^4$ cells per well. The chamber was placed in a 24-well plate, and its membrane was coated with Matrigel (100 μg/cm$^2$) (BD Bios, Franklin Lakes, NJ, USA). U87 MG cells were cultured in the upper compartment, and 10% FBS-containing medium was added to the lower compartment. After 24 h, the chambers were washed twice with PBS, fixed in 4% paraformaldehyde for 15 min and washed. Then, the cells were stained with 0.5% crystal violet for 15 min and washed twice. Photography was conducted with a microscope (ECLIPSE Ni-U, Nikon, Tokyo, Japan), and the number of cells per field was counted for the shNT group and the sh-Loxl1-1 & 2 groups.

## Statistical methods

Statistical distinctions between the two cohorts were assessed through Student's unpaired two-sided t test. Within the arms of multiple groups, one-way analysis of variance (ANOVA) was used, accompanied by subsequent *post hoc* least significant difference assessments for multiple pairwise comparisons. Survival evaluations, encompassing both Kaplan–Meier survival analyses and univariate/multivariate Cox regression analyses, were performed with SPSS (Chicago, IL, USA) and R software (*R Core Team, 2020*). A significance threshold of $P < 0.05$ was consistently upheld as the standard for statistical significance across all analytical appraisals.

## RESULTS

### Genomic sequencing of LOXs and prognostic variant identification

To delineate the correlation between genetic aberrations in LOXs and the prognostic landscape of GBM patients, we analyzed a cohort comprising 429 specimens obtained from three distinct genomic profiling studies of GBM patients. The LOXs family, which includes all five constituents, exhibited varying degrees of genetic perturbations within the GBM patient pool. The cumulative frequency of such genetic alterations reached 2.6%, accounting for 11 of the 429 samples under scrutiny. As shown in Figs. 1A and 1B, 0.4% of GBM patients presented missense mutations in Lox. This mutation often results in Lox dysfunction, with 269 and 274 amino acid changes (aa), and affects the phosphorylation and O-linked glycosylation of Lox. Missense mutations and inframe mutations were detected in Loxl1 with the same rate of genetic alteration as Lox. This missense mutation led to 421 amino acid changes in Loxl1. Loxl2 was the gene with the most genetic alterations (0.8%), and its mutation may contribute to changes in gene phosphorylation. A total of 0.4% of GBM samples presented missense mutations, which caused 545 amino acid changes in Loxl3. A total of 0.69% of GBM samples showed a missense mutation and amplification in Loxl4, of which 583 amino acid changes were detected in Loxl4.

To further determine the prognostic value of LOXs with genetic alterations in GBM, we performed a Kaplan–Meier analysis of 11 GBM patients with LOXs genetic alterations from three GBM cohorts (Fig. 1C). GBM patients exhibiting genetic modifications pertaining to LOXs displayed notably truncated overall survival intervals compared to those of their counterparts devoid of such alterations (Fig. 1D). The prognostic genetic alteration of LOXs in GBM indicated the important clinical value of LOXs.

### LOXs expression analysis and correlated biological functions

We performed differential expression analyses of LOXs in TCGA and CGGA at the transcriptional level, as shown in Figs. 2A and 2B. In contrast to their expression levels in normal tissue counterparts, the expression profiles of all LOXs (Lox, Loxl1, Loxl2, Loxl3, Loxl4) were notably elevated within the domain of GBM tissues, indicating the plausible oncogenic functions of LOXs in GBM. To confirm the functional involvement of LOXs in GBM, we subjected the 250 most relevant genes (50 for each respective member) to comprehensive scrutiny through GO analyses. The processes of extracellular structure organization (biological process, BP) and extracellular matrix structural constituent

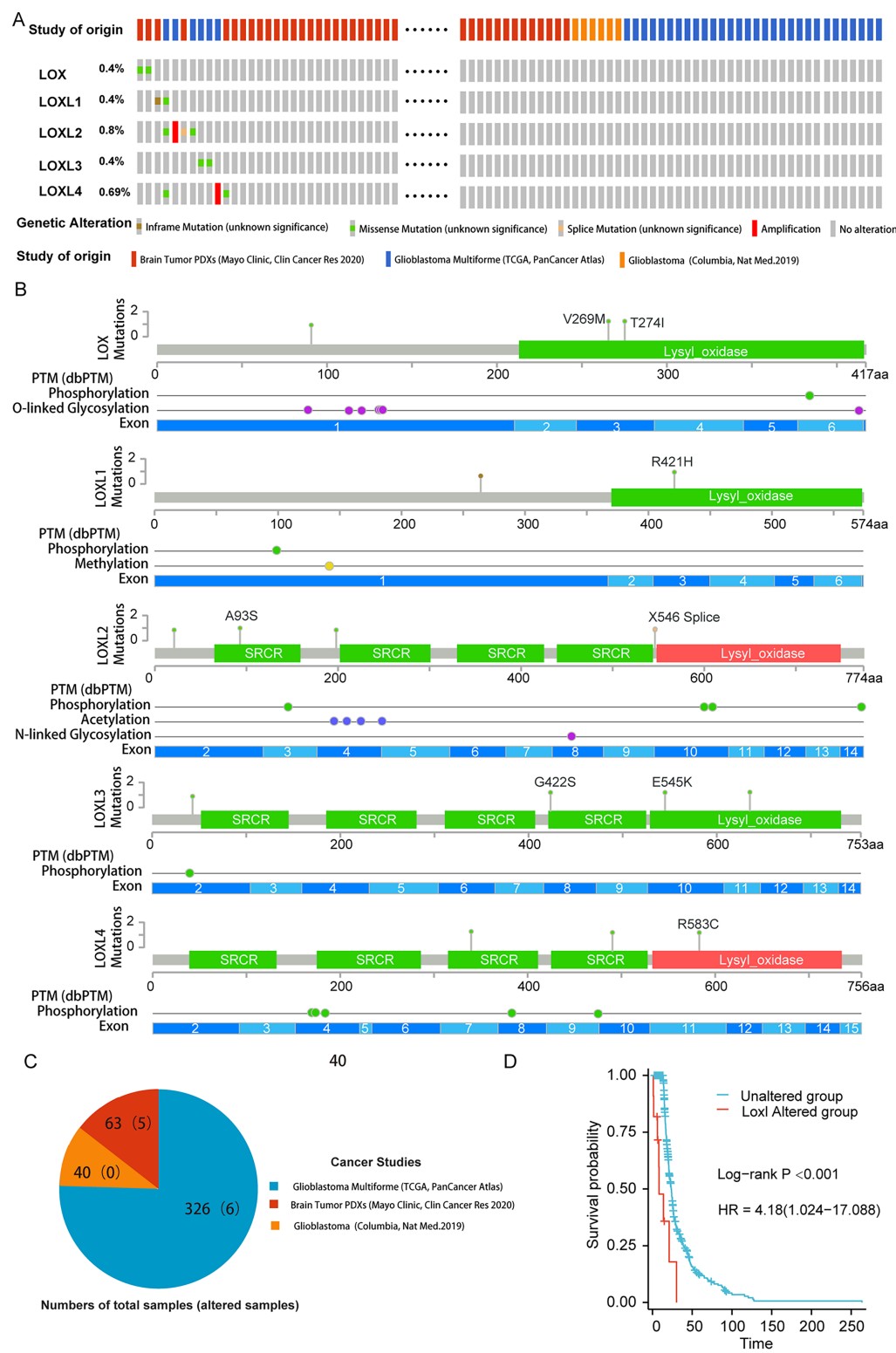

**Figure 1 Genetic mutation and prognosis of LOXs in GBM.** (A) Amplification and deep deletions are presented in red and blue boxes, while the remaining boxes indicate no or no alterations from cBioPortal. (B) Related modifications and corresponding exons associated with LOX alterations in GBM. SRCR: Scavenger receptor cysteine-rich domain. (C) The samples with altered LOXs in cancer studies are shown

**Figure 1** (continued)
in the pie diagram. The total sample number of each study was out of parentheses and the number of LOXs altered samples are shown in parentheses. (D) Genetic modifications in LOXs among GBM patients were subjected to scrutiny utilizing Kaplan–Meier analysis. The unaltered cohort is represented by the blue dashed line, which is juxtaposed by the red dashed line indicating the cohort with genetic alterations.

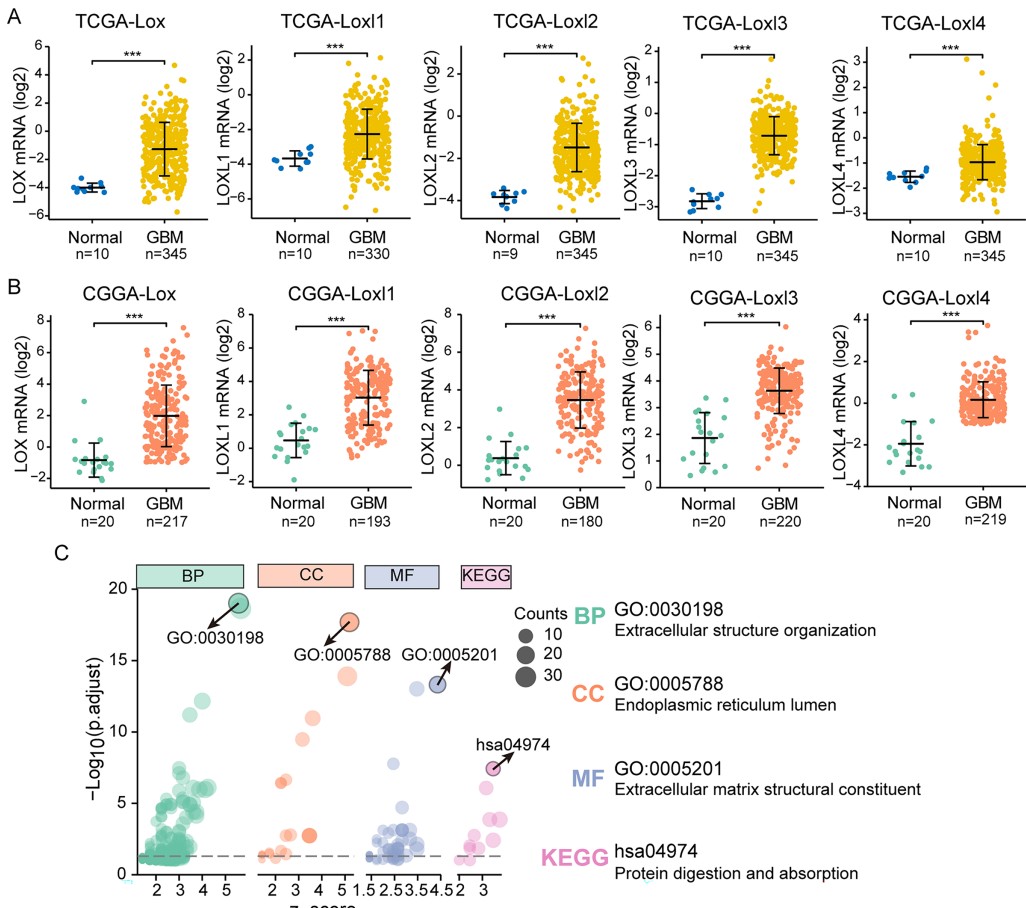

**Figure 2 Differential mRNA expression and related pathways of LOXs in GBM.** Aberrant mRNA expression of LOXs in GBM from TCGA (A) and CGGA (B), in contrast to normal tissues. (***$P < 0.001$). (C) The top 50*5 genes related to LOXs were included for GO and KEGG analyses. The GO analyses revealed biological process (BP), cellular component (CC), and molecular function (MF) terms. The most strongly correlated results are indicated with arrows, and their annotations are listed.

(molecular function, MF) in the endoplasmic reticulum lumen (cellular component, CC) were correlated with the expression of LOXs in GBM based on the GO and KEGG analyses. Moreover, KEGG analysis revealed that the biological roles of LOXs in GBM were associated with the pathways of protein digestion and absorption (Fig. 2C). Together, these results suggested that LOXs serve as important regulators of ECM protein expression.

## LOXs and clinical GBM classifications

GBM is classified into the proneural, classical, and mesenchymal subtypes. The classifications were related to patient survival, of which mesenchymal subtypes displayed the shortest median survival (*Verhaak et al., 2010*). As shown in Figs 3A, 3B, we found that, except for Loxl1 in the CGGA, all LOXs in GBM were more highly expressed in the mesenchymal subtypes than in the classical and proneural subtypes, indicating that LOXs could serve as diagnostic indicators for predicting mesenchymal subtypes in GBM patients.

Another important indicator widely used in GBM diagnosis and treatment is IDH mutation (*Yan et al., 2009*). It is associated with better outcomes in IDH1-mutant patients than in IDH1-WT patients. We next explored the correlation between IDH mutation status and LOXs expression. Interestingly, we found that all LOXs in GBM were more highly expressed in the IDH1-WT subgroup than in the IDH1-mutant subgroup (Figs. 3C, 3D). The correlation between high LOX expression and IDH1-WT expression was confirmed both in the TCGA and CGGA cohorts, suggesting that LOXs could act as promising markers in GBM.

## Independent prognostic variables and nomogram construction

To determine the effective prognostic factors of LOXs in GBM patients, two separate databases were used. Overall survival of GBM patients with high Lox, Loxl2, Loxl3, and Loxl4 expression was similar to that of patients with low Loxl2, Loxl3, and Loxl4 expression. Moreover, patients with higher Loxl1 levels might have a shorter survival time, which might be a prognostic indicator for clinical diagnosis and treatment (Fig. 4A).

The elucidation and validation of prognostic indicators in GBM encompassed an intricate interplay between patient-specific clinical attributes and Loxl1 expression patterns. Parameters spanning age, sex, tumor classifications encompassing IDH mutation status and distinct GBM subtypes, as well as the therapeutic modalities of chemotherapy and radiotherapy, were critically engaged in this multifaceted endeavor. The results of univariate Cox survival analyses revealed that younger age, IDH mutation status, chemotherapy, radiotherapy, and low Loxl1 expression were protective variables, which translated into an extended survival period for patients with GBM (Fig. 4B).

To ascertain the clinical predictive efficacy inherent to Loxl1 expression, we integrated the aforementioned clinical prognosticators into a multivariate Cox survival analysis framework. Through the deliberate exclusion of sex and GBM subtypes as confounding elements, we proceeded to gauge the adjusted hazard ratios for these prognostic determinants along with their corresponding 95% confidence intervals (Fig. 4C). To ascertain the clinical predictive efficacy inherent to Loxl1 expression, we integrated the aforementioned clinical prognosticators into a multivariate Cox survival analysis framework. Through the deliberate exclusion of sex and GBM subtypes as confounding elements, we proceeded to gauge the adjusted hazard ratios for these prognostic determinants along with their corresponding 95% confidence intervals (Fig. 4D). The nomogram included age, IDH mutation status, chemotherapy, radiotherapy, and Loxl1 expression as pivotal parameters contributing to its predictive power.

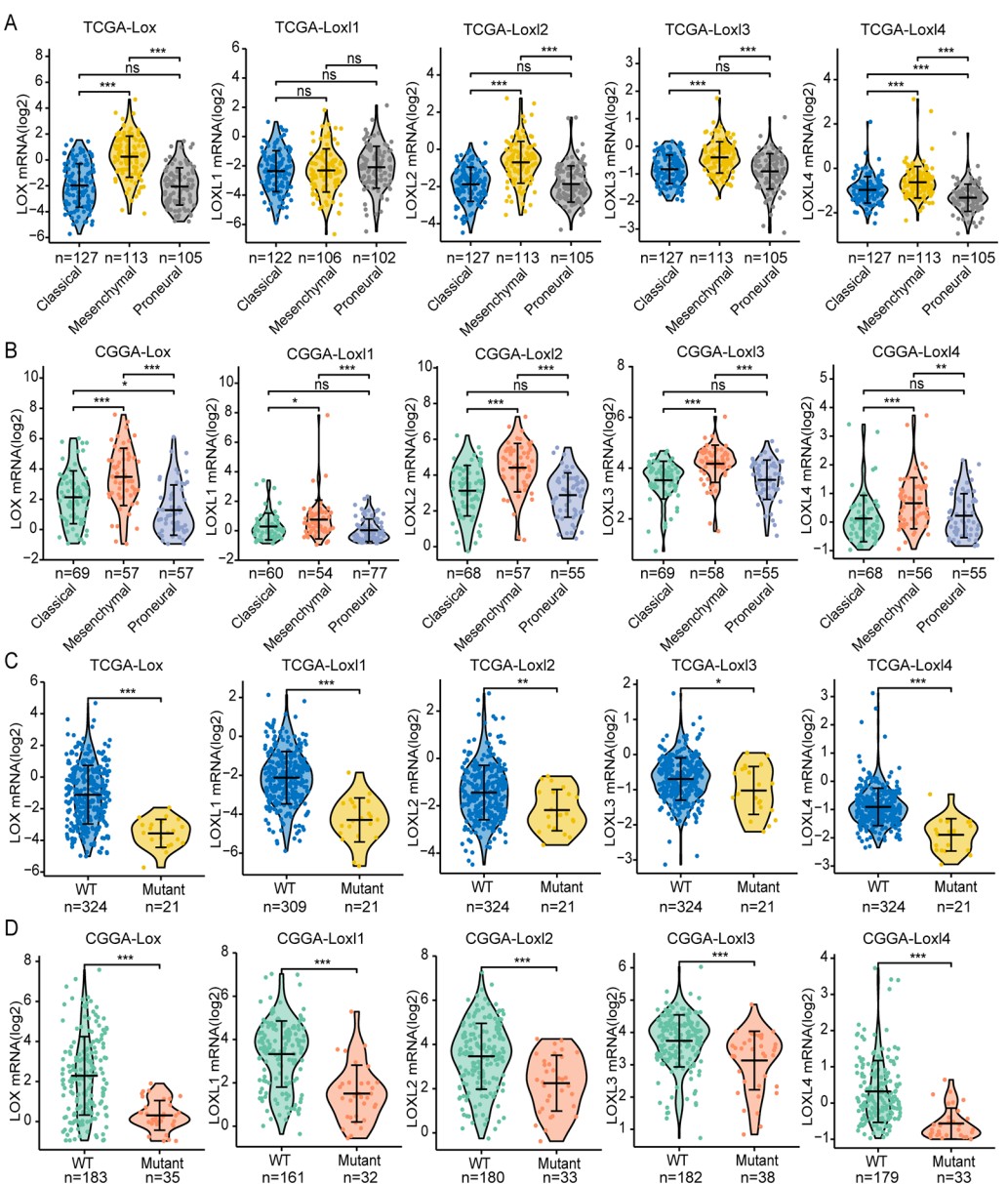

**Figure 3 Differential mRNA expression of LOXs in GBM molecular subtypes and IDH mutation.** Distinct mRNA expression of LOXs was observed across classical, mesenchymal, and proneural subtypes (TCGA, A, and CGGA, B), and IDH1-WT/mutant groups (TCGA, C and CGGA, D). (*$P < 0.05$, **$P < 0.01$, ***$P < 0.001$).

## Verification and evaluation of the Loxl1-based nomogram in GBM cohorts

Within the training cohort of GBM patients from the TCGA, the nomogram and PRSM were constructed based on the combination of Loxl1 and clinicopathological parameters, including age, IDH mutation status, chemotherapy, and radiotherapy (Fig. 5A). The computation of individual patient risk scores facilitated an insightful determination, where elevated risk scores exhibited a noteworthy correlation with diminished survival intervals

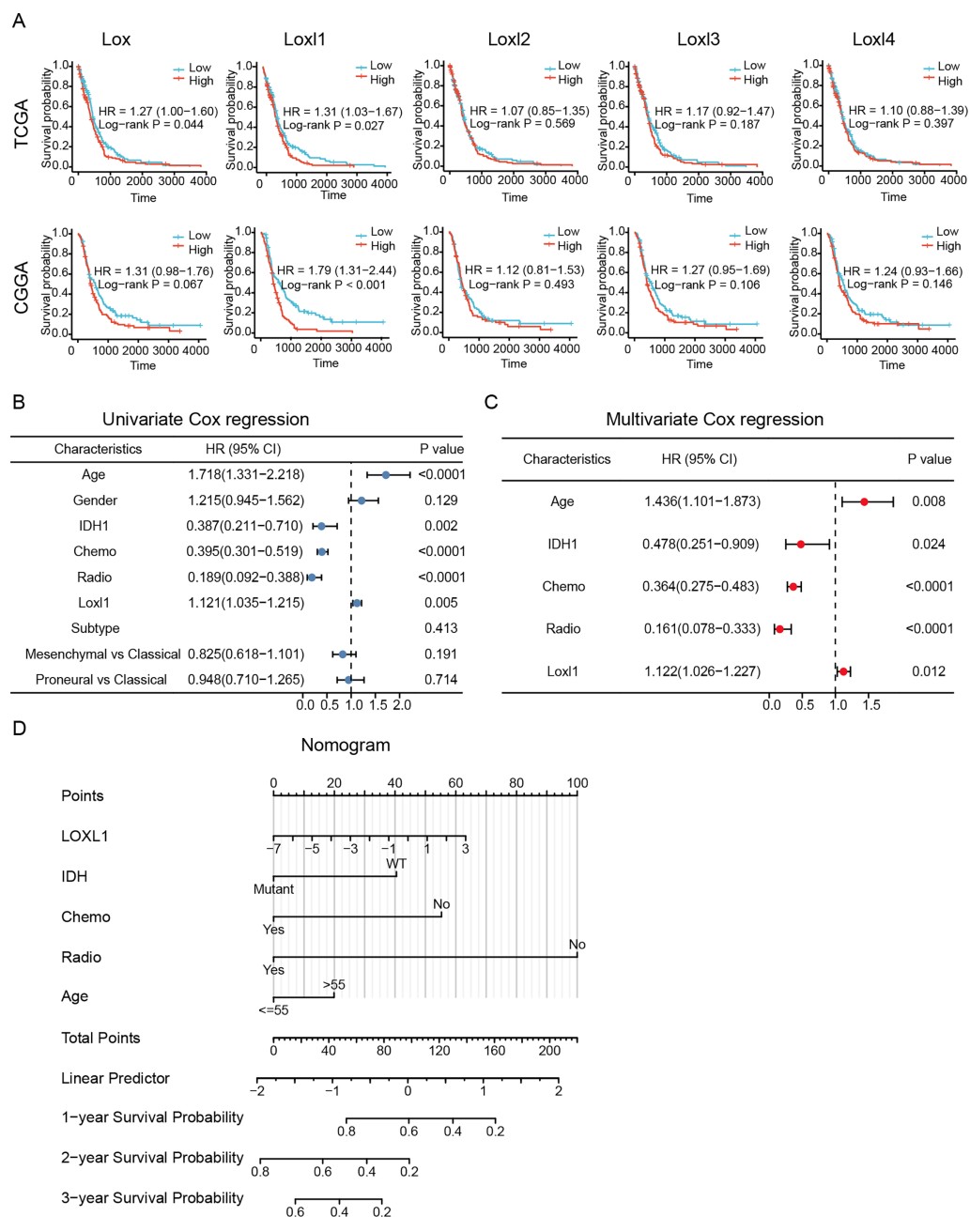

**Figure 4 Survival analysis and Loxl1-based nomogram and PRSM construction.** Kaplan-Meier analyses (A). Further univariate (B) and multivariate (C) Cox survival analyses were utilized to confirm the protective variables in GBM. (D) Prognostic factors based on the nomogram and the PRSM were constructed, and specific points of individual patients were assigned to predict survival time.

(HR = 1.82–2.97, *P* < 0.001) (Fig. 5D). To demonstrate the generalizability of the nomogram model, the CGGA GBM cohort and another different TCGA GBM cohort were utilized to confirm the model's predictive efficiency. The risk scores of both validation sets were also calculated and are presented in Figs. 5B, 5C. Kaplan–Meier survival analyses of the CGGA and external TCGA validation cohorts revealed results similar to those of the

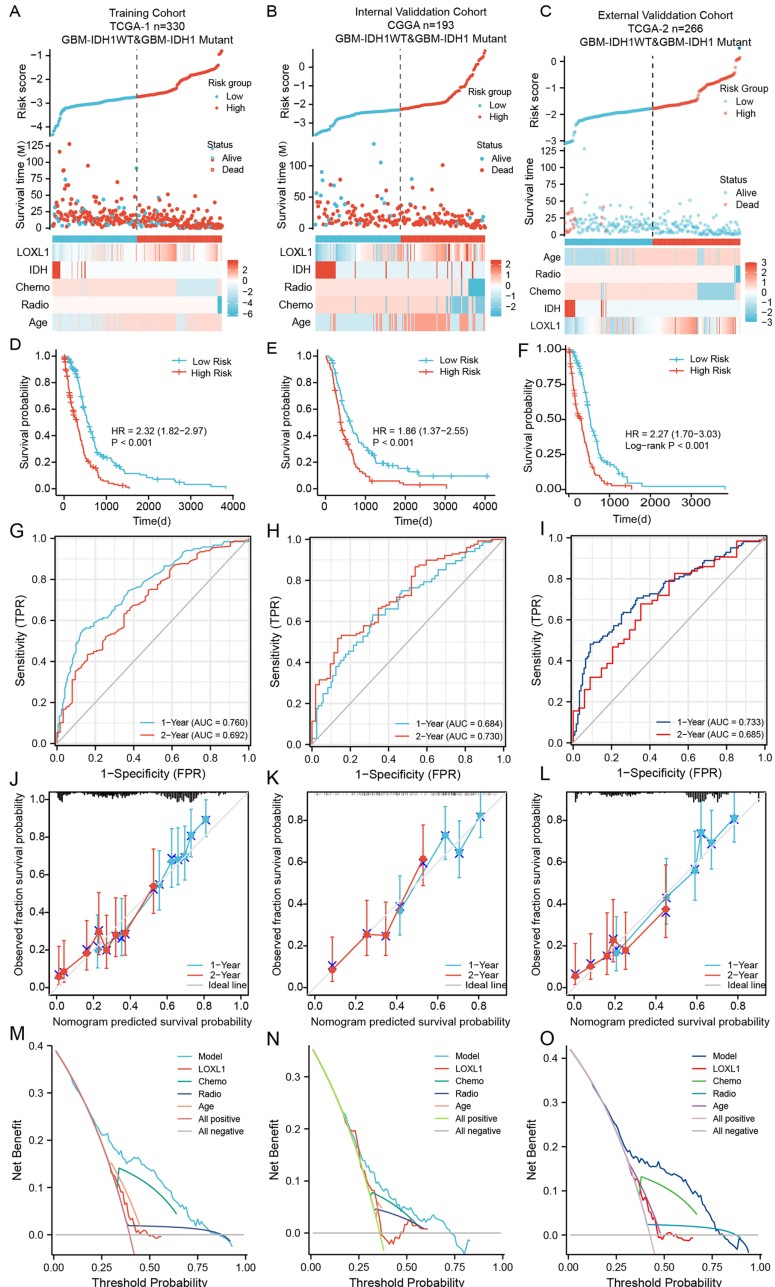

**Figure 5** **The distribution and evaluation of Loxl1-based nomogram model.** Risk scores were calculated by the nomogram (top), while the corresponding survival time and clinicopathological factors are also shown in the dot plot (middle) and heatmaps (below) from the training TCGA (A), CGGA (B), and external TCGA (C) cohorts. The examination of associations between risk scores and survival duration among GBM patients included investigations in the training TCGA cohort (D), the CGGA cohort (E), and an independent external TCGA dataset (F). The ability of the model to predict 1- and 2-year survival times is shown as the area under the curve (AUC) (for the training TCGA (G), CGGA (H), and external TCGA (I) cohorts). The 1-year prediction is shown in blue, and the 2-year prediction is shown in red. The values that the model provided, and the actual observations are both presented in the calibration plots. The 1-year prediction is visually represented in blue, while the 2-year prediction is distinguished by a red hue, encompassing datasets from the training (J), CGGA (K), and external TCGA (L) datasets. The different net clinical benefits that patients obtained from the nomogram and clinical features were calculated and presented by decision curve analysis (DCA) (the training (M), CGGA (N), and external TCGA (O) sets).

training TCGA cohort (Figs. 5E, 5F), suggesting that the Loxl1-based PRSM classification functions stably in different populations.

To substantiate the prognostic precision of the model, we calculated the area under the curve (AUC) through a comprehensive analysis of t-ROC curves. Our findings indicate that all cohorts exhibited notably substantial AUC values. Specifically, the AUC values for the Loxl1-based nomogram for predicting 1- and 2-year overall survival durations were 0.760 and 0.792, respectively, in the TCGA training set, 0.684 and 0.730, respectively, in the CGGA validation set, 0.733 and 0.685, respectively, in the external TCGA validation set (Figs. 5G–5I), suggesting that the Loxl1-based nomogram and the PRSM were highly accurate and robust. We also generated a calibration curve of the nomogram in different cohorts. The diagrams show a congruence between the actual and projected outcomes at the 1- and 2-year time points concerning the survival duration of GBM patients, which was validated across both the training cohort (Fig. 5J) and the independent validation cohorts (Figs. 5K, 5L). The clinical applicability of the established Loxl1-based nomogram was evaluated by decision curve analysis (DCA). More net benefits could be achieved by the nomogram for almost all threshold probabilities when compared with other prognostic variables in the training (Fig. 5M) and validation cohorts (Figs. 5N, 5O). These results suggested that the Loxl1-based PRSM had a better ability to predict survival than other variables in primary GBM patients.

Cox regression analysis of the training cohort notably revealed a statistically significant association between the presence of IDH mutations and overall survival duration among GBM patients, including those with IDH1- mutations and IDH1-WT. However, the 2021 WHO classification further defined patients with IDH1-WT as GBM patients (*Louis et al., 2021*). We further excluded IDH-mutant patients from all cohorts and utilized the Loxl1-based nomogram without the IDH mutation parameter. The risk scores were calculated again (Figs. S1A–S1C), and the high-score groups had shorter survival times (Figs. S1D–S1F) in both the training and validation sets. Based on the new t-ROC (Figs. S1G–S1I), calibration curve (Figs. S1J–S1L), and DCA (Figs. S1M–S1O) analyses, we found that the prediction accuracy and net benefits of the model were still stable regardless of whether GBM was characterized by IDH1 mutation.

## Loxl1-related enrichment pathways and tumor immune infiltrates in GBM

As shown in Fig. 6A, we processed the GSE182109 scRNA-seq data, and the distribution of all the raw cell samples is shown. The expression of Loxl1 in glioma cells of GBM tissues could help us understand its functional state. The stringent quality control metrics previously described were used to screen 29,402 high-quality cells. These cells were scaled and clustered into 13 clusters (Fig. 6B). Based on the annotated marker genes (Table S6), we classified the cells into myeloid cells (C0, C3), glioma cells (C1, C2, C5, C7, C8, C10, C11), T cells (C4), pericytes (C9), endothelial cells (C12), and oligodendrocytes (C6) (Fig. 6C). The glioma cells were separated and are shown in Fig. 6D according to tumor grade (LGG or GBM). Combined with visualization of the Loxl1 expression pattern (Fig. 6E), we found that Loxl1 was highly expressed in the GBM groups.

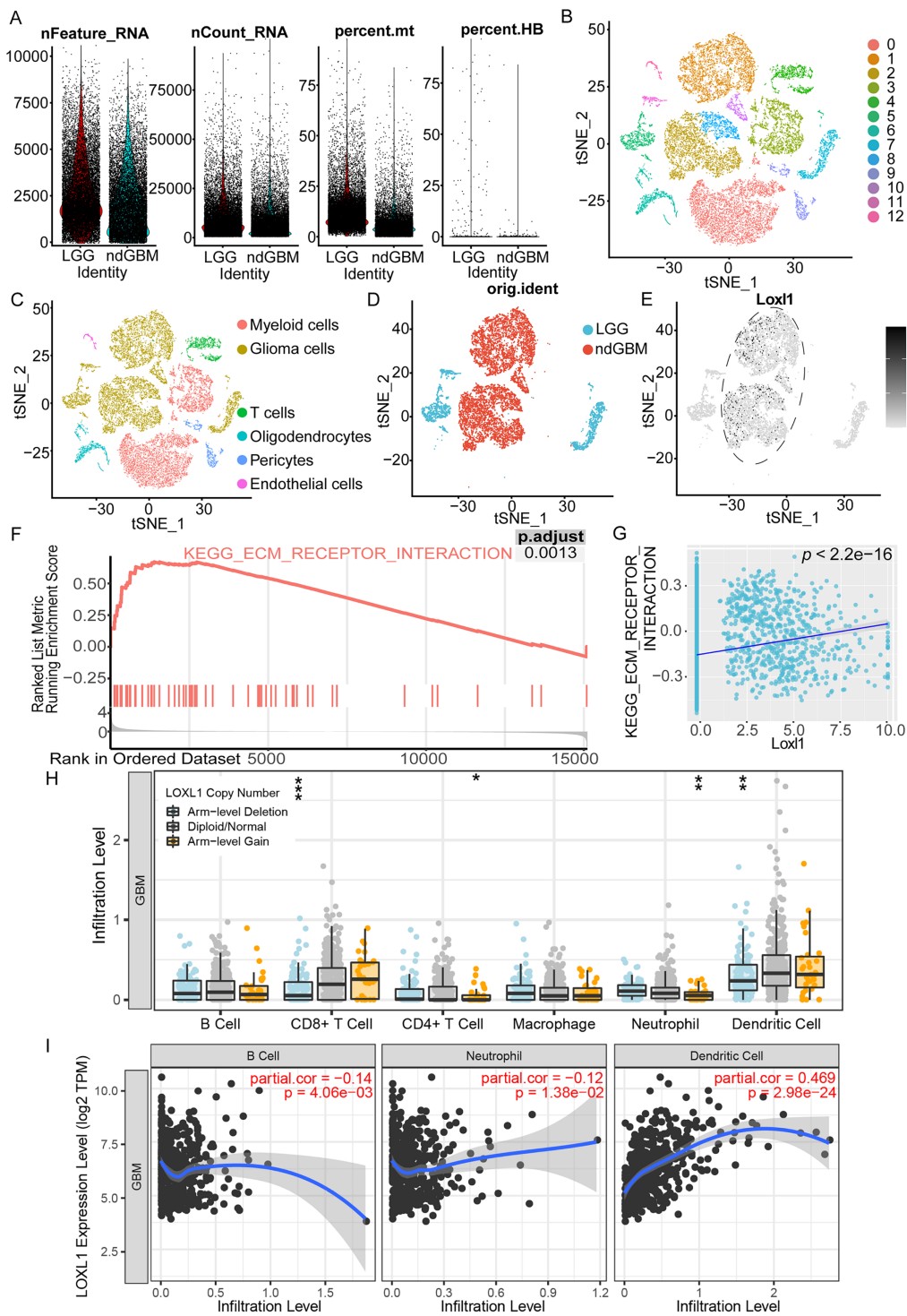

**Figure 6 Functional enrichment pathway and tumor immune infiltration analyses of Loxl1 in GBM.**
(A) The quantification of distinct RNA attributes, including nFeature_RNA, nCount_RNA, the relative extent of mitochondrial content (percent.mt), and the proportion of hemoglobin genes (percent.HB), was undertaken for each individual cell subjected to sequencing within the GSE182109 dataset. (B) Quality control filtered cells clustered *via* the tSNE dimensionality reduction algorithm and all cells were clustered into 13 clusters (C0-12). (C) The clusters were further annotated by specific marker genes, including myeloid cells (C0, C3), glioma cells (C1, C2, C5, C7, C8, C10, C11), T cells (C4), pericytes (C9),

**Figure 6** (continued)
endothelial cells (C12), oligodendrocytes (C6). (D) Glioma cells were separated and grouped by tumor grades (blue represents LGG, and red represents primary GBM). (E) The expression of Loxl1 (black dots) was mainly in the dashed circle area (GBM groups). (F) Enriched pathways in GBM were analyzed by gene set enrichment analysis: ECM receptor interaction (red lines, adjusted $P = 0.0013$). (G) The correlation between Loxl1 and enriched pathways in GBM analyzed by gene set variation analysis: ECM receptor interaction ($P < 0.05$). (H) With gain (orange dots) or deletion (blue dots) of Loxl1 CNA (copy number alteration) levels, the infiltration levels of B cells, CD8+ T cells, CD4+ T cells, Neutrophil, and dendritic cells in GBM were changed. The grey dots meant normal CNA groups. (I) With the mRNA level of Loxl1 increasing, the infiltration levels of B cells, neutrophils, and dendritic cells were also significantly changed. The *P-value* and correlation rates were exhibited. (*$P < 0.05$, **$P < 0.01$, ***$P < 0.001$).

To explore the enriched biological functions in GBM, GSEA was employed, and the results revealed that ECM receptor interaction ($P < 0.05$) was positively associated with GBM (Fig. 6F). Furthermore, GSVA was performed to calculate the pathway scores of each cell. The expression of Loxl1 was also found to be positively correlated with the interaction of ECM receptors (Fig. 6G). Previous results of GO and KEGG analyses of GBM bulk sequencing data support the above results, suggesting that Loxl1 might participate in the progression of tumor invasion. We also explored the correlation between Loxl1 and various immune cells. Based on the CNA levels, we found that arm-level gain of Loxl1 was significantly associated with CD8+ T cells and dendritic cells infiltration, while its arm-level deletion was correlated to CD4+ T cells and neutrophils infiltration (Fig. 6H). At the mRNA level, Loxl1 expression was positively related to dendritic cells and negatively correlated with B cells and neutrophils (Fig. 6I), which implies that Loxl1 has promising potential for assessing the efficacy of GBM immunotherapy.

## Decreased Loxl1 inhibited the invasion of GBM cells

Loxl1-knockdown U87 MG cells were generated to further confirm the invasive role of Loxl1 in GBM (Figs. 7A, 7B). CCK-8 assays revealed that the viability and proliferation of cells in the sh-Loxl1 group were significantly lower than those in the control group (Fig. 7C). The EMT pathway plays vital roles in tumor cell invasion and migration. As previously reported, the expression of E-cadherin, N-cadherin, Vimentin and Snai1 is associated with the EMT pathway. First, we found that N-cadherin was decreased (Figs. 7D, 7E) and that E-cadherin was increased (Figs. 7F, 7G) in both sh-Loxl1 groups. Decreases in Vimentin and Snai1 were also observed by western blot analyses following the downregulation of Loxl1 (Figs. 7H, 7I; Figs. 7J, 7K). To determine the effect of Loxl1 on the motility and invasion of U87 MG cells, wound healing and Transwell assays were performed. Wound closure was lower in the sh-Loxl1 group than in the control group (Figs. 7L, 7M). The number of invasive glioma cells in the Transwell assay also decreased significantly after shLoxl1 knockdown (Figs. 7N, 7O), confirming that Loxl1 contributes to the invasion of glioma cells. These data confirmed that Loxl1 plays an invasive role in tumor progression by regulating the EMT pathway, which might be a promising therapeutic strategy for GBM patients.

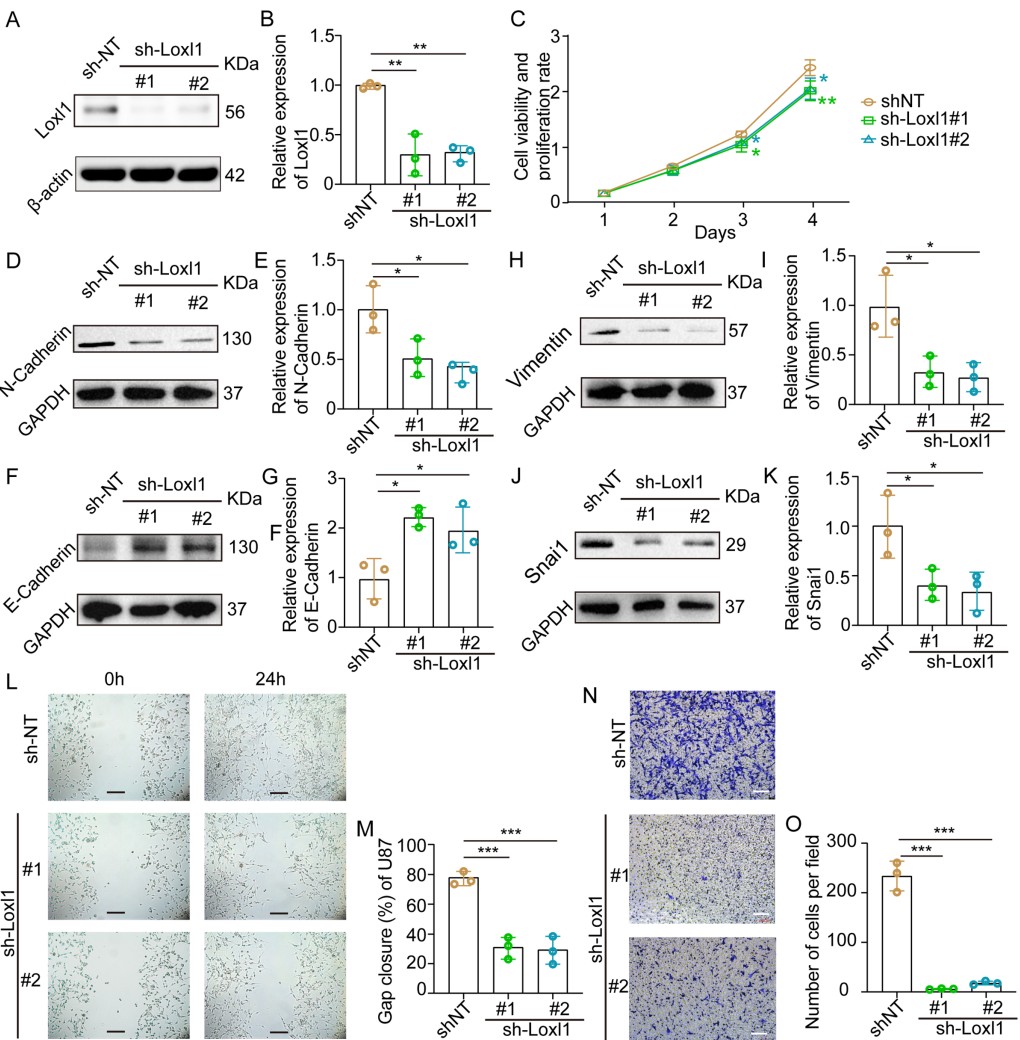

**Figure 7 Downregulation of Loxl1 suppressed tumor invasion in GBM.** (A, B) The downregulation of
Loxl1 by specific lentiviral vectors. $N = 3$. (**$P < 0.01$). (C) The viability and proliferation of cells in shNT
(orange curve), shLoxl1-1 (green curve) and shLoxl1-2 (blue curve) groups were assessed by CCK-8
assay. $N = 5$. (*$P < 0.05$, **$P < 0.01$). The expression of the invasion-related proteins N-cadherin (D, E),
E-cadherin (F, G), Vimentin (H, I) and Snai1 (J, K) was detected after treatment with Loxl1. $N = 3$.
(*$P < 0.05$, **$P < 0.01$). (L, M) The wound healing assay in Loxl1-knockdown glioma cells at 0 and 24 h.
The scale bar represents 200 μm. $N = 3$. (***$P < 0.001$). (N, O) Transwell invasion assays were also
performed to validate the relationship between Loxl1 expression and glioma invasion. The bar represents
200 μm. $N = 3$. (***$P < 0.001$).             

## DISCUSSION

LOXs, which are pivotal enzymes orchestrating ECM protein dynamics within the TME,
have been implicated in steering tumor progression (*Laurentino et al., 2022*). Nevertheless,
the comprehensive importance and practical utility of LOXs in the prognostic appraisal of
primary GBM patients have yet to be exhaustively elucidated. Our research revealed that
GBM patients with altered LOXs had shorter survival times. All LOXs displayed high
mRNA expression in two separate GBM datasets. In particular, we confirmed that high

Loxl1 promoted tumor invasion with prognostic value. Based on the outcomes derived from Kaplan‒Meier and Cox regression analyses, an innovative nomogram and PRSM predicated on Loxl1 were formulated and subsequently subjected to rigorous validation procedures, providing a highly accurate and practical tool for GBM prognosis assessment in clinical practice. Additionally, we explored the roles of Loxl1 in tumor immune infiltrates and first validated the invasive role of Loxl1 in the progression of GBM, revealing potential intervention targets for glioma treatment.

Our investigation discerned the prognostic significance inherent in genetic alterations of LOXs in GBM. Remarkably, we provided inaugural confirmation that GBM patients harboring genetic alterations in LOXs exhibited notably shorter survival times. LOXs were also reported to regulate collagens in the tumor ECM (*Kim et al., 2022*), which was similar to our GO and KEGG analyses in that LOXs were closely related to the regulation of ECM proteins in GBM. A poor prognosis was found for GBM patients with high Lox expression in the TCGA cohort ($P = 0.044$) but not in the CGGA cohort ($P = 0.067$). These results were partially consistent with those of previous studies (*Han et al., 2014*; *Huang et al., 2018*; *Patnam et al., 2022*; *Xia et al., 2022*). They reported the independent prognostic role of Lox in lower-grade glioma, suggesting that the prognostic value of Lox is related to the grade of glioma. In addition, a recent study in 51 GBM patients (*Zhang et al., 2022*) confirmed the prognostic value of Lox. Similar to our results, these data demonstrated that different clinical variables and sizes of GBM cohorts might result in different prognostic effects of Lox in GBM patients, which needs to be proven in a larger and complete GBM cohort.

The expression of Loxl1 is significantly related to the antiapoptotic effects and migration of glioma cells (*Li et al., 2018*; *Yu et al., 2020*). Consistent with our results, previous studies have indicated a similar prognostic role for Loxl1 in glioma using a single Kaplan‒Meier analysis (*Xia et al., 2022*; *Yu et al., 2020*). In this study, we further included more clinicopathological factors and performed further Cox regression analyses in multiple GBM cohorts. Genetic alterations and transcriptional changes in Loxl1 were significantly associated with the tumor immune microenvironment. Moreover, based on our scRNA-seq analysis, we confirmed that Loxl1 expression was closely related to glioma invasion *in vitro*. The EMT pathway in GBM was also inhibited by the downregulation of Loxl1. To the best of our knowledge, this study represents an inaugural report of the prognostic significance of Loxl1 in GBM patients. Moreover, the results corroborated the invasive capacity of Loxl1 in GBM. Together, these data indicated that Loxl1, an invasion-related indicator, might facilitate clinical prognostic evaluation and personalized treatment of GBM patients.

Loxl2 is correlated with increased tumor dimensions and elevated tumor grade in glioma (*Du & Zhu, 2018*). Previous studies in glioma indicated the independent prognostic role of Loxl2 (*Du & Zhu, 2018*; *Xia et al., 2022*). Loxl3 can promote tumor cell invasion in GBM (*Laurentino et al., 2021*), and Loxl4 is recognized as a stemness-related prognostic biomarker in glioma (*Lvu et al., 2020*). These results partially differed from our results in GBM cohorts. Although discernible upregulation of Loxl2, Loxl3, and Loxl4 was detected

in GBM tissue, the prognostic relevance of these genes for GBM patients within the TCGA and CGGA cohorts was not demonstrated. These findings collectively underscore the potential contributory role of Loxl2, Loxl3, and Loxl4 in glioma progression. However, they also emphasize that these molecules may not necessarily serve as optimal indicators for prognostic evaluation among GBM patients.

GBM heterogeneity and individual asymmetrical clinical features influence patient prognosis, and therefore, we constructed a novel nomogram based on individual clinicopathological factors and the mRNA expression of Loxl1. The Loxl1-based nomogram exhibited good accuracy, reliability, and clinical benefits in multiple GBM cohorts. Compared to previous GBM prediction models (*Zheng et al., 2022*) and deep machine learning models (*Poirion et al., 2021*), the performance of our nomogram and the PRSM was acceptable and practical. In addition, the public databases in our study were built before 2021, and GBMs were defined as IDH1-WT and IDH1-mutant. According to the new 2021 WHO CNS tumor classification for GBM (*Louis et al., 2021*), we excluded IDH1-mutant patients from three GBM cohorts and further assessed the Loxl1-based model in IDH1-WT GBM cohorts. Our results confirmed that the nomogram stably and extensively predicted IDH1-WT GBM patient survival and is a novel, stable, and individualized tool for assessing GBM patient prognosis.

Nevertheless, our study has several limitations that need to be clarified. First, this was a population-based retrospective study, and selection bias was inevitable even though the inclusion and exclusion criteria were strict. Second, although three large populations of GBM cohorts were included in our study, the number of normal samples in our study was smaller than the number of GBM samples. More population cohorts worldwide are needed to further validate the above findings and improve the application of the nomogram. Third, using single-cell analyses, we found that Loxl1 contributed to tumor invasion in GBM and conducted experiments *in vitro*. The detailed regulatory mechanism and interactions with Loxl1 in GBM tumor microenvironments should be addressed in future studies. Overall, our investigation comprehensively evaluated the distinct contributions of each constituent of LOXs to the prognosis of GBM patients. Moreover, we constructed a pragmatic nomogram derived from an amalgamation of multiomics and multidatabase analyses, serving as an invaluable tool for both prognostic appraisal and tailoring of individualized therapeutic approaches for GBM patients.

## CONCLUSIONS

In conclusion, we found prognostic genomic and transcriptional changes in LOXs in GBM. IDH1-wild-type and mesenchymal (not Loxl1) GBM subtypes also exhibited upregulation of LOXs. Loxl1 plays an independent prognostic role in GBM by participating in tumor immune infiltrates and promoting tumor invasion *via* the EMT pathway. A novel Loxl1-based nomogram and PRSM were constructed and exhibited high accuracy in predicting GBM patient survival time, reliability, and net clinical benefits in clinical GBM prognosis assessment.

### Funding
The authors received no funding for this work.

### Competing Interests
The authors declare that they have no competing interests.

### Author Contributions
- Gui-Qiang Yuan conceived and designed the experiments, performed the experiments, analyzed the data, prepared figures and/or tables, authored or reviewed drafts of the article, and approved the final draft.
- Guoguo Zhang analyzed the data, prepared figures and/or tables, and approved the final draft.
- Qianqian Nie analyzed the data, prepared figures and/or tables, and approved the final draft.
- Zhong Wang conceived and designed the experiments, authored or reviewed drafts of the article, and approved the final draft.
- Hong-Zhi Gao conceived and designed the experiments, performed the experiments, authored or reviewed drafts of the article, and approved the final draft.
- Gui-Shan Jin conceived and designed the experiments, authored or reviewed drafts of the article, and approved the final draft.
- Zong-Qing Zheng conceived and designed the experiments, performed the experiments, analyzed the data, prepared figures and/or tables, authored or reviewed drafts of the article, and approved the final draft.

### Data Availability
The TCGA GBM RNA-seq data is available at GlioVis data portal: http://gliovis.bioinfo.cnio.es/; Dataset: TCGA GBM.

The CGGA RNA-seq data is available at http://www.cgga.org.cn; mRNAseq_693 and mRNAseq_325.

The single-cell RNA-seq datasheet is available at GEO database: GSE182109.

The raw measurements are available in the Supplemental Tables.

### Supplemental Information
Supplemental information for this article can be found online at http://dx.doi.org/10.7717/peerj.17579#supplemental-information.

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
