# Peer review of "Lysyl oxidase-like 1 predicts the prognosis of patients with primary glioblastoma and promotes tumor invasion via EMT pathway"

_PeerJ, doi:10.7717/peerj.17579_

## Round 0.1 · original submission · Major Revisions

Please address all the queries of all reviewers and amend the manuscript accordingly.

**Language Note:** The review process has identified that the English language must be improved. PeerJ can provide language editing services - please contact us at [email protected] for pricing (be sure to provide your manuscript number and title). Alternatively, you should make your own arrangements to improve the language quality and provide details in your response letter. – PeerJ Staff

Reviewer 1 ·

Basic reporting

The manuscript has several grammatical errors. It is difficult to read. I consider that it is essential to be reviewed in depth by a language expert or a professional editing service.

The study describes an interesting use of LoxL1 as a prognostic value in patients with GBM, but I consider that it is not sufficiently consistent and original.

Experimental design

The introduction is poorly detailed, especially the physiological role that LOXs have.

In the materials and methods section there is little detail on the cell culture procedures, lentivirus generation, selection of cells transduced with the shLoxl1 lentivirus, western blot, Wound healing assay and Transwell invasion assay. It is essential to indicate whether the Wound healing assay tests have been done in culture medium with or without FBS. It is necessary to analyze whether LOXl1 interference affects the survival and proliferation of U-87 MG cells. I recommend analyzing the EMT pathway in more depth, studying other markers to reach a more solid conclusion.

I consider it necessary to indicate in the materials and methods section the meaning of the p value, in addition to the figures.

Validity of the findings

In the discussion, the description of the limitations of the study is positively valued.

The description of the figures need to be significantly improved. Numerous errors (example: figure 1D – unchanged cohort in green but it is blue; figure 2: the number of samples analyzed in each group is not indicated, the number of normal tissue samples is much lower than that of GBM samples, figure 1C it is difficult to interpret)

Figure 6 presents numerous errors. Errors in the figure caption description and in the main text where "figure 6C" is indicated several times (lines 335-338). Furthermore, the sentence is incomplete, demonstrating little care in reviewing the article before sending it for publication.

Additional comments

In conclusion, I consider that the manuscript does not meet the appropriate scientific requirements to be published in the journal. The conclusions of the manuscript are too ambitious, it is necessary to further analyze the role that LoxL1 has in the EMT transition.

Reviewer 2 ·

Basic reporting

This paper studied the role of lysyl oxidase-like 1 in glioblastoma. The authors found that Lox1 was related to tumor invasion and this was validated by three more assays. Overal, this article is well-written. The figures and tables are of good quality.

I have two major concerns.

First, the reasons for studying Loxl1 in this study weren't explained very well in the second paragraph of the Introduction. The authors listed many publications that studied Loxl1, and the necessity and novelty of this paper need to be added in the introduction.

Second, the Materials and Methods section is written in an unusual style and it is hard for other researcher to repeat their experiment.

Experimental design

No comment

Validity of the findings

The data that the authors provided are robust and statistically sound.

Reviewer 3 ·

Basic reporting

no comment

Experimental design

no comment

Validity of the findings

no comment

Additional comments

The finding of this article is interesting. The prognostic factors of LOXL1 gene were demonstrated in GBM, and various data analysis methods were used to validate the argument. Nevertheless, minor revision is needed before consideration for acceptance:
1. Line 155-156: There are double "CD8+ T cell",should it be "CD8+ T cell, CD4+ T cells"?
2. Line 170-171: anti-Loxl1(#XXX?, 1:500, Santa Cruz Biotechnology). Please add product number of the antibody.
3. U87MG and U87 cells should be unified in the article.
4. Line 179: "... by Image J" should be added with "(NIH, Bethesda, MD, USA)", Line 179 : "(NIH, Bethesda, MD, USA)" should be deleted.
5. Fig 7. legend: "(A,B),(G,H)" should be bold according to requirements.
6. "P-value" in Fig 6. legend should be italic.
7. Formatting errors exist in some of the references, such as missing page numbers.
8. What‘s the relationship between the expression of IDH and LOXL1? Is it positive or negative correlated?

---

## Round 0.2 · Minor Revisions

Please address the remaining issues pointed out by reviewer #1 and amend your manuscript accordingly.

Reviewer 1 ·

Basic reporting

The quality of the article has significantly improved. The quality of the English language is much superior to the previous version, correcting all grammatical errors; now the text is perfectly understandable.

Experimental design

The introduction is now more comprehensive.
The materials and methods section has been notably improved and completed.

Validity of the findings

However, there are still several things to correct:

The addition of the sample number (n) included in each studied group in Figures 2A and 2B is positively valued, so it would be appreciated if the 'n' were also included in all the images of Figure 3.
In Figure 4A, Loxl1, the p-value is not included in TCGA.
In Figure 6, I believe there is an error in the description of Figure 7H: gain (YELLOW dots) or deletion (BLUE dots). Include the significance of 'p' in the figure description (***, **, *).
Regarding Figure 7, I have several comments:
The addition of viability and proliferation studies in Figure 7C is appreciated. The analysis of Vimentin and Snai1 is positively valued.
In this figure, there are data from the western blot images that do not correspond to the quantification shown in the figure, specifically in Figures 7H and 7I, where it can be seen that the intensity of the vimentin signal from the sh-Loxl1#1 sample is notably lower than that of sh-Loxl1#2, however, the quantification shows that they are practically equal. Additionally, the original Vimentin blots are not provided.
In the provided original N-Cadherin blots, the differences in signal between sh-NT and the sh-Loxl1 samples #1 and #2 are not as evident as those shown in image 7E, which indicates that there would be half the signal between both groups.

Additional comments

No comment.

Reviewer 2 ·

Basic reporting

I am pleased to note that all of my previous comments have been thoroughly addressed. I have no further suggestions and recommend the paper for publication.

Experimental design

no comment

Validity of the findings

no comment

Additional comments

no comment

Reviewer 3 ·

Basic reporting

no comment

Experimental design

no comment

Validity of the findings

no comment

Additional comments

no comment

---

## Round 0.3 · accepted · Accept

All required changes have been made, and the revised manuscript is acceptable now

Reviewer 1 ·

Basic reporting

All the proposed modifications and corrections have been made, so the article is ready to be published.

Experimental design

No comments.

Validity of the findings

No comments.

Additional comments

No comments.